# Vulnerable, Resilient, or Both? A Qualitative Study of Adaptation Resources and Behaviors to Heat Waves and Health Outcomes of Low-Income Residents of Urban Heat Islands

**DOI:** 10.3390/ijerph191711090

**Published:** 2022-09-04

**Authors:** Lawrence A. Palinkas, Michael S. Hurlburt, Cecilia Fernandez, Jessenia De Leon, Kexin Yu, Erika Salinas, Erika Garcia, Jill Johnston, Md. Mostafijur Rahman, Sam J. Silva, Rob S. McConnell

**Affiliations:** 1Suzanne Dworak-Peck School of Social Work, University of Southern California, Los Angeles, CA 90089, USA; 2Department of Population and Public Health Sciences, Keck School of Medicine, University of Southern California, Los Angeles, CA 90032, USA; 3Department of Earth Sciences, Dornsife College of Letters, Arts and Sciences, University of Southern California, Los Angeles, CA 90089, USA

**Keywords:** health equity, heat waves, climate change, adaptation behaviors, health impacts, urban heat islands

## Abstract

Little is known of how low-income residents of urban heat islands engage their knowledge, attitudes, behaviors, and resources to mitigate the health impacts of heat waves. In this qualitative study, we conducted semi-structured interviews with 40 adults in two such neighborhoods in Los Angeles California to explore their adaptation resources and behaviors, the impacts of heat waves on physical and mental health, and threat assessments of future heat waves. Eighty percent of participants received advanced warning of heat waves from television news and social media. The most common resource was air conditioning (AC) units or fans. However, one-third of participants lacked AC, and many of those with AC engaged in limited use due primarily to the high cost of electricity. Adaptation behaviors include staying hydrated, remaining indoors or going to cooler locations, reducing energy usage, and consuming certain foods and drinks. Most of the participants reported some physical or mental health problem or symptom during heat waves, suggesting vulnerability to heat waves. Almost all participants asserted that heat waves were likely to increase in frequency and intensity with adverse health effects for vulnerable populations. Despite limited resources, low-income residents of urban heat islands utilize a wide range of behaviors to minimize the severity of health impacts, suggesting they are both vulnerable and resilient to heat waves.

## 1. Introduction

According to the most recent report of the Intergovernmental Panel on Climate Change, the frequency, intensity, and duration of extreme heat events are projected to continue to increase through the twenty-first century, with all regions expected to experience unprecedented temperatures [1]. These events have been associated with increased heat-related morbidity and mortality [2,3,4,5,6,7]. Rising temperatures have been directly connected to human health through heat-related illnesses (e.g., heat exhaustion, heat syncope, and heat stroke), resulting in a marked increase in hospital admissions and mortality rates, although these increases are influenced by variations in exposure, location, and susceptibility [8]. For instance, low-income residents of urban heat islands are more susceptible to heat-related morbidity and mortality due to increased temperatures, lack of tree cover and green space, low socioeconomic status, lack of material resources, and low saturation of air conditioning (AC) [9,10,11,12,13,14]. Heat waves and rising temperatures have also been linked to increased negative emotions and stress [14,15,16], hospitalizations for mood and behavioral disorders [17], mortality in patients with mental and behavioral health disorders [18,19], and increased rates of suicide [20,21,22,23] and homicide [24,25].

To prevent the occurrence of these adverse impacts, increased attention has been devoted to reducing social and environmental vulnerability [3,26,27] and to building, supporting, and sustaining individual and community resilience [28,29,30]. Resilience is the ability of an individual or community to resist, absorb, accommodate, adapt to, transform, and recover from the effects of persistent stress or a disruptive event in a timely and efficient manner [31,32]. Resilience is also related to the concept of adaptive capacity, defined as “the ability of systems, institutions, humans and other organisms to adjust to potential damage, to take advantage of opportunities, or to respond to consequences” ([33], p. 542). Adaptive capacity is often linked to vulnerability; however, determining the natures of the relationships involving these two constructs is complicated by differences in how they are defined in the literature and how they are linked together [34]. For instance, in a risk-hazard approach, vulnerability is determined by adaptive capacity, whereas in a contextual approach, vulnerability determines adaptive capacity [35,36]. Interpreting vulnerability as an outcome leads to a focus on mitigation of greenhouse gas emissions, technical adaptations, and compensation for climate change. In contrast, interpreting vulnerability as contextual results in a focus on addressing existing social and economic marginalization and inequities and adopts social adaptation and sustainable development as the solution [35].

Social and environmental vulnerability to heat waves has been studied extensively [3,5,37,38], especially as it relates to risk perception [37,39,40,41,42,43]. Empirical studies have found that low-income individuals are less likely than higher-income individuals to be aware of heat warnings but more likely to be concerned that heat would make them ill [37,39,40,41,42,43]. Low-income individuals also less likely to have AC due to its expense, a lack of perceived need, or personal dislike [40,41,42]. They are less likely to engage in positive adaptive behaviors due to limited resources [40]. However, this research has provided us with only a limited perspective on the adaptive capacity of vulnerable populations. As described by Martin and Paneque ([44], p. 2),


*“Adaptive capacity is a common component of vulnerability and resilience assessments, depending on how these concepts are framed, but the isolated measuring and modeling of adaptive capacity is not as advanced as the broader concepts (e.g., vulnerability, resilience). Although several scholars are trying to frame and quantify this ability of adapting to actual and future events, there is still very little consensus as to what constitutes effective and adaptive adaptation, and therefore little agreement about models and indicators to measure it”.*


In reaching such agreement, one of the persistent questions to be addressed concerns the relationship between vulnerability, viewed as a barrier to adaptive capacity, and resilience, viewed as a facilitator to adaptive capacity.

Given the absence of a widely accepted model or theory of adaptive capacity and behaviors, qualitative methods are recommended to explore the parameters and potential causal linkages underlying a phenomenon in order to develop a model that will lead to the generation and testing of hypotheses using quantitative methods [45]. Earlier research has not examined in depth the lived experience of adapting to heat waves of those who are most vulnerable, i.e., low-income residents of urban heat islands [11]. Two qualitative studies [43,46] identified adaptive behaviors but only those of older adults. An understanding of this experience is critical to identifying resource needs and developing better assessments of adaptive capacity and interventions targeting this population. Qualitative research is also needed to better understand racial/ethnic and socioeconomic disparities in adaptive resources (e.g., AC) and behaviors and sharpen interventions to address root causes [37].

To address this lack of information, we conducted a qualitative study of adaptation resources and behaviors of a group of adults living in two urban heat islands in Los Angeles, California. Qualitative methods were used to explore issues that have not been examined in previous research. Our aims were to explore the following questions: (1) How do low-income residents of urban heat islands respond to heat waves; i.e., what adaptive behaviors (adaptation strategies) do they implement during a heat wave? (2) What resources (adaptation capacity) are available to these residents to enable them to adapt to heat waves, i.e., vulnerability and resilience? (3) How are vulnerability and resilience associated with adaptive capacity? (4) How have such events and the adaptation behaviors and resources implemented in response to them affected the physical and mental health of these residents, i.e., health outcomes?

## 2. Materials and Methods

The authors followed the Standards for Reporting Qualitative Research (SRQR) to report the findings from this study [47] (see Appendix A).

### 2.1. Design

In this study, we operated from a conceptual framework that highlights both the social determinants of health, emphasizes community strengths as well as vulnerabilities, and offers guidance on the development and implementation of community selected programs, policies, and interventions. This framework draws from the Integrated Climate Change and Health Indicator Systems Framework [48] that combines elements of the WHO’s Drivers-Pressures-State-Exposure-Effect-Action model [49] and the Multiple Exposures–Multiple Effects framework [50], placing multi-step causal chains associated with climate change within a context of socioeconomic and demographic factors, societal actions, and other non-climate drivers.

### 2.2. Participants

Participants for the proposed project were a purposively selected sample [51] of parents of middle and high school students participating in the University of Southern California’s (USC) McMorrow Neighborhood Academic Initiative (NAI), the university’s signature college prep program, which enrolls close to 1000 students annually. USC McMorrow NAI is an academically rigorous and comprehensive, seven-year pre-college program designed to prepare students from South and East Los Angeles for admission to a college or university. Under the program guidelines, students must commit to a seven-year plan of attending Saturday Academy classes along with weekday morning classes at USC, after-school tutoring, and parent workshops. Parents are also required to attend a biweekly Family Development Institute program to create a 360-degree, hands-on approach to reinforce student academic goals and study habits and maximize a healthy home environment. Students must meet the following selection criteria: (1) a resident from the target neighborhoods; (2) eligible to attend one of three neighborhood schools; (3) a 5th grade student getting ready to enter the 6th grade; (4) most students have a C+ average or better; and (5) students must agree to attend all Saturday Academy sessions for seven years, and parents must agree to attend the family development institute on Saturdays. A presentation was made in English, Spanish, and Mandarin to all parents attending a Family Development Institute meeting in November 2021. To be eligible for participation in the study, parents or caregivers were required to live or work in one of the neighborhoods with schools attended by NAI students and to have lived in the neighborhood for a minimum of 3 years. Of the 225 parents attending the virtual meeting, 66 (29.3%) signed up to participate in the interviews. Members of this subsample were contacted and scheduled to be interviewed until theoretical saturation was reached (i.e., no new information was obtained from participants) [52]. The final sample of participants included 40 parents representing 40 different households. Each parent received a USD 30 Visa or Amazon gift card as compensation for their participation.

All but two of the participants resided in two Los Angeles inner city neighborhoods as defined by census tracts [53]. One neighborhood had an average temperature that was 4C above the average for the city, 55.2% impervious surface, 3% tree cover, and 99.1% minority population. The other neighborhood had an average temperature that was 1.7C above the average for the city, 79% impervious surface, 7.4% tree cover, and 96.2% minority population. The remaining two participants also resided in urban heat islands but in different parts of the city.

### 2.3. Data Collection

Using a semi-structured interview guide we developed that was based on the Integrated Climate Change and Health Indicator Systems Framework [46], all interviews were conducted online using the Zoom platform or by telephone. Bilingual research assistants conducted interviews in Spanish and Mandarin with non-native English-speaking participants. Participants provided information on the following: 

Measures of proximate exposure: past experience with heat waves (“Have you ever experienced a heat wave that seemed to last for a long time?”); assessment of threats associated with heat waves (e.g., “Do you think there have been more or fewer heat waves now than there were, say 10 years ago? If you think there have been more, can you think of any reason why that is the case? Do you think these heat waves create any physical or mental health problems for people? Do you ever worry about the health of your children during a heat wave?”)

Measures of adaption strategies: knowledge of methods for preventing heat-related illness (e.g., “Is there anything you can or should do to prepare for heat waves before they occur?”); typical use of AC during the summer months; ways used or recommended for keeping cool and adapting to heat waves (e.g., “Did you do anything to keep cool or stay comfortable during this time? Based on your knowledge and experience, is there anything you might recommend to people like yourself on how to prepare for or respond to heat waves or days of poor air quality?”)

Measures of vulnerability and resilience: demographic characteristics; place and type of residence; access to and type of AC; health status (e.g., “Do you or anyone in your family have any health conditions that would make you more susceptible to heat or air pollution-related illness (e.g., respiratory problems, heart problems, asthma)? Did anyone check in on you, either in person or by phone, to see how you were doing?”)

Measures of health outcomes: self-reported impact of heat waves on personal physical and mental health and health of children (e.g., “Did the heat have any noticeable effect on your health or well-being or the health and well-being of anyone in your family?”)

Interviews were conducted between late November 2021 and early January 2022 and lasted between forty-five minutes and one hour, and all but two interviews were digitally recorded. Interviewers summarized participant responses to questions throughout the interview to ensure their accuracy and validity. All study activities were reviewed and approved by the Institutional Review Board of the University of Southern California.

Information on neighborhood economic and environmental characteristics were obtained from publicly available data sets. This included median household income data by zip codes, available from the U.S. Census [54]; characteristics of urban heat islands, available from ESRI [55]; and urban heat island index data by census tract, available from the California Environmental Protection Agency [56].

### 2.4. Data Analysis

Digital records were transcribed and checked for accuracy by interviewers. Transcripts of interviews conducted in Mandarin or Spanish were translated into English using the Sonix AI web-based services and verified for accuracy by bilingual research assistants. Interview transcripts were analyzed using a thematic content analysis methodology [57]. Transcripts were distributed among and reviewed by investigators to develop a broad understanding of content related to the project’s aims and to identify topics for discussion and observation. Short descriptive statements or “memos” were prepared to document initial impressions of topics and themes and their relationships and to define the boundaries of specific codes (i.e., the inclusion and exclusion criteria for assigning a specific code) [58].

Transcripts were then independently coded to condense the data into analyzable units. Segments of text ranging from a phrase to several paragraphs were assigned open codes based on a priori (i.e., from the interview guide) or emergent themes [57]. Axial codes were assigned to describe connections between categories and between categories and subcategories [59]. Codes were also assigned to reflect the social and demographic characteristics of study participants. Lists of codes developed by each investigator were matched and integrated into a single codebook.

Three transcripts were independently coded by four members of the research team. Disagreements in assignment or description of codes were resolved through discussion between investigators and by refining definitions of codes. The final list of codes or codebook, constructed through team consensus, consisted of a list of themes, issues, accounts of behaviors, and opinions that relate to knowledge, attitudes, and adaptive behaviors during heat waves. With the final coding structure, two investigators separately reviewed three transcripts to determine level of agreement in the codes. An average level of agreement of 93.4% indicated excellent reliability in qualitative research [60]. Based on these codes, the computer program NVivo20 was used to generate a hierarchical arrangement series of categories. These groups of codes were used to further the process of axial or pattern coding to examine the association between different a priori and emergent categories. Finally, by constantly comparing these categories with each other, the different categories were further condensed into broad themes [57]. Consensus in responses to questions was estimated based on the number of participants who provided a specific answer.

## 3. Results

### 3.1. Participant Demographics

The demographic characteristics of the 40 study participants are listed in Table 1 below. All but one of the participants were female, with an average age of 42 years. Two-thirds of the participants were Latinx; one in five were Asian-American. Sixty-two percent had a high school education or less. More than half were employed outside of the home, and the majority rented their place of residence. Participants resided in their current neighborhoods an average of 15 years. These neighborhoods had an average median household income of USD 49,358 and an average urban heat index of 7229.

Every parent who participated in an interview could recall having experienced a heat wave since living in their current place of residence. Analysis of the interview transcripts revealed four themes: (1) an assess of the risks of similar heat waves and their consequences in the future; (2) adaptation resources; (3) behaviors intended to avoid excessive heat or keep cool; and (4) mental and physical health impacts of these events. Each of these themes are described in detail below.

### 3.2. Threat Assessment of Future Heat Waves

Thirty-eight of the forty participants (95%) asserted that there were more heat waves now than there were 10 years ago (Table 2). One participant reported she did not know, and one participant did not think it was hotter now than 10 years ago. The most commonly cited reasons for the increase in temperatures generally and frequency of heat waves were environmental contaminants and insufficient efforts at recycling (n = 15), global warming and the greenhouse effect (n = 10), deforestation (n = 10), air pollution (n = 6), lack of concern for the environment (n = 6), and overpopulation (n = 5). Other factors responsible that were cited by participants included wildfires (n = 3), dependence on fossil fuels (n = 2), and the constant demand for new technology (n = 2).

The majority of participants expressed concern that the continued increase in temperature and frequency of heat waves would lead to adverse physical and mental health effects. As explained by one participant, the connection between high temperatures and air pollution was of particular concern:


*“Yes, yes. That’s why I’m constantly watching, because I know that the hotter it gets, the more pollutants. And although we do not perceive them, we go up here on a level. We have CO_2_, nitrogen, in other words, pollutants that can affect our health. That’s why I’m also looking at air quality, because whether it’s hot or cold, it’s going to influence pollutants. And since we go outside, we can be more exposed.”*


Thirteen of the participants were especially concerned about the health of their children, especially when they were attending schools with no clear policies for protecting them during heat waves.

### 3.3. Heat Wave Adaptation Resources

Some of the resources that study participants used in preparation for heat waves are listed in Table 1. The most common resource used by study participants during heat waves was AC units or fans. Twenty-five participants (62.5%) were living in homes or apartments with central AC, window units, or mini-splits (i.e., ductless wall units). However, 15 participants (37.5%) reported not having AC units either because landlords placed restrictions on their installation in rental properties (n = 3), it was too expensive to purchase (n = 6), the cost of electricity required to power such units was too excessive (n = 2), or it was unnecessary (n = 3). Three participants reported that use of AC was bad for their health. Use of portable or ceiling fans, especially in the bedrooms, were reported by 24 (60%) participants.


*“They don’t want us to do any kinds of arrangements, or, you know, make any holes on the walls anything like that, so we just avoid any issues.”*



*“Maybe because I’m from Guangdung, the heat in our region is relatively humid, and the whole body is sweaty when it’s hot. Because it feels like the heat here is better.”*



*“Uh, we don’t usually like to use air conditioners, because it’s not good for our health. In fact, we only turn it on when we feel very hot, usually before going to bed.”*


Another resource used by some of the participants was their social networks. These included family members, friends, and neighbors. Ten of the participants (25%) reported having been in contact with a relative, friend, or neighbor during a visit or by telephone during the last heat wave. As reported by one of the older participants, *“My daughters, you know they call in and check on us um ‘What are you doing,’ you know, whatever, and, you know, try not to go out. You know, all that kind of thing, you know? It used to be like we were lecturing them, but now they’re lecturing us.”* One participant reported checking on an older family member: *“We did check on our on my husband’s parents because they live on the lower level. And it gets the same, the same heat from the house is super hard. So, we then checked on them many times.”* In some instances, family members would check up on one another:


*“I have relatives in the inland empire, so it tends, it kind of tends to get a little bit hotter than here. And we will check up on each other, you know talk about the weather hot, it is when we’ve been waiting stay cool if they have central AC you know what sometimes I think that one night we actually stayed over their house, because it was much, much cooler than here. Yeah, so that’s, that’s pretty much what we would do as far as checking in on each other.”*


Other residents reported checking up on neighbors or contacting neighbors for help in the event of a medical emergency.

Although twenty-one participants (52.5%) reported no such contact, eight participants indicated that their knowledge of how to prepare for and respond to heat waves came from social interactions with family, friends, or health care providers. One participant noted:


*“I guess growing up in the city and in school, like having adults, you know, hearing my parents and hearing other adults say, you know, say we’re staying inside because of the air quality or because it’s hot. Don’t exert yourself. Don’t overdo anything right now and, I think I’ve just learned by, I guess, by like the directions of adults growing up.”*


A third resource was the availability of greenspace in the community. Eight participants mentioned having access to parks during heat waves:


*“Oh, we were going out to the park. We went to the park a lot. We also used to go out like this. Eh? Where there is a place that my husband drive us to? He drives farther than I do. There’s a place you can ride bicycles and we bring food. We took a sheet with us so we could lie down on the grass and be there, that is, in the open air. I think that sometimes it was cooler under a tree than in the house. So what we used to do was that I would take them with their backpacks. I tell them to take their backpacks to do their homework, because they will concentrate more there. In other words, we are all going to do homework together and we can be there as a family, enjoying a snack. I would bring sandwiches and I would take what I could. In the park as well we tried to go where there weren’t too many people just because sometimes there would be many people and also a lot of noise. My husband knows several parks, so we would go under the biggest trees, with the biggest shade to be able to be quiet, to enjoy, to be calm and that is what helped us a lot.”*


Knowledge also constituted an important resource for study participants. The most common source of information on how to adapt to heat waves came from reading or watching the news on television (n = 14), personal experience of adapting to previous heat waves and high temperatures (n = 10), or school (n = 6):


*“When I was with my child’s dad, he was working in construction, and he suffered a heat stroke. He had a heatstroke and he practically woke up in the sun on his job because he works outdoors. And he got a heatstroke that day and it gave him a headache and everything. And that’s why we know they have to drink a lot of water or if they suffer from heatstroke, they have to take a lot of Pedialyte and rest to recover.”*



*“I’m in the parent center at my children’s school. So there’s like first aid, the parents who are on the board give us like, ah, like basic information about what we have to do, like when we are helping as volunteers in the school, in the playgrounds or in the dining area or on vacations.”*


Finally, most (80%) of the participants reported that they had been alerted to a forthcoming heatwave through the television or social media. In some instances, parents received text messages for their children’s schools, advising them to prepare for a heat wave by making certain their children were appropriately dressed and well hydrated:


*“I only heard it from the news. But I don’t exactly know if it was from the news or the phone about the weather. That’s what I’ve been hearing on the radio. It’s going to be very hot or something.”*



*“Yes, from the news we get to watch at night. It’s how we inform ourselves.”*



*“Yes, because they put it on the platforms: Facebook or Twitter, Instagram, Snapchat, the newscasts announce it by time, that it’s going to be a very big heat wave, fires or it’s going to be very strong air and all that. What is it called? Expands more the heat or it’s dry area, pure steam like, like, like desert-like? No, dry?”*


### 3.4. Heat Wave Adaptation Behaviors

The most common behavior reported during a heat wave was drinking more water. This was reported by all 40 participants (Table 3). This was followed by the use of fans and portable AC units during heat waves (n = 25). These devices were used primarily in late afternoon and evening hours when everyone in the family was at home or when the outdoor temperature reached a specific level (90–95° (n = 2)). As noted earlier, the need to cool indoor temperatures by using AC units was often counterbalanced by concerns about high electricity bills, safety, or power outages. Changes in clothing were also reported by several participants. This included wearing lighter clothes (n = 15) and/or hats (n = 4) when outdoors.

Another adaptation behavior revolved around nutritional practices. Several participants reported eating fresh foods that do not require cooking (n = 7) or consuming items that are cold such as ice cream or cold snacks (n = 6). Chinese and Spanish-speaking participants reported using culturally specific methods for coping with extreme heat, including consuming particular types of teas, soup, and fruits and vegetables believed to cool the body down (n = 8).

Household modifications was another commonly adopted practice during a heat wave and included closing windows, curtains, or blinds (n = 8) and opening windows when temperatures began to drop (n = 8); limiting use of appliances that generate heat (n = 5); and planting trees on the property (n = 4). Other changes included insulating the attic (n = 2), cleaning the house (n = 2) or AC unit (n = 1), and turning off the lights to save energy (n = 2).

Another common practice employed during heat waves revolved around location. Although many participants reported avoiding going outside during a heat wave (n = 15), others reported going elsewhere, including outdoor swimming pools (n = 4), parks with trees (n = 3), other places with shade (n = 3), public places with AC (n = 5), or visiting a relative who has AC (n = 1). Only one participant reported going to a specially designated community cooling center.

Other practices for staying cool during a heatwave reported by participants included taking cold showers (n = 10), wrapping oneself in cold towels or clothes (n = 4), freezing bags of ice (n = 1), staging water balloon games outdoors for children (n = 2), using the AC in the car (n = 3), sleeping in the living room with fans on (n = 2), refraining from exercising indoors (n = 1), and use of umbrellas (n = 1) and sunscreen (n = 2) when outdoors. However, in general, participants reported engaging in several different behaviors when adapting to heat waves:


*“We for the kids, we just did like, you know, try to do like water balloons when it wasn’t too hot outside. We wouldn’t go out during the middle of the day. Then that was just, you know, trying to stay cool inside and wearing a short-sleeved clothes or shorts, trying to keep the doors closed when it was, you know, trying to keep the cool air in and then windows open at night, trying to get some ventilation in. But I mean, other than that, we were trying to go like into the mall. We tried to do like water activities, but no other than that, no.”*


### 3.5. Health Effects of Heat Waves

Participants reported several different types of physical and mental health impacts experienced during heat waves (Table 4). More than half (57.5%) of the participants reported some health problem or physical discomfort during heat waves, including fatigue or lethargy (n = 7), circulatory (headaches, swollen feet) (n = 6) and respiratory (n = 2) issues, heat rashes (n = 4), dry lips or throats (n = 3), or loss of appetite (n = 2). Participants also reported their children experienced nose bleeds (n = 3), feeling flushed (n = 1), headaches (n = 1), heat rashes (n = 2), or allergies (n = 2). Almost three-fourths of the participants (n = 29) reported changes in affect during heat waves, including anxiety (n = 19), depression (n = 7), and irritability (n = 3). Two participants reported that their children experienced anxiety during heat waves, and one participant reported their children being irritable. However, except for prescribed medications for dermatological conditions, none of the participants reported having to seek medical attention for these conditions.

## 4. Discussion

The four themes identified from the interview transcripts in this study reflect the role of knowledge (a resource), attitudes (perceived risk), and behavior revealed in four major components of an integrated model of heat wave adaptation, illustrated in Figure 1: risk of heatwaves, responses or adaptive behaviors, resources of adaptive capacity, and results of health impacts of heat waves. The relationships between and among these components are described below:

### 4.1. Risk of Heatwaves

Our findings revealed a near universal acknowledgement among residents of low-income neighborhoods in Los Angeles of the growing frequency and severity of heat waves in Southern California. In their assessment of the threat of future heatwaves, all but two of our forty participants (95%) asserted that there were more heat waves of greater intensity than there were ten years ago, and that this trend was likely to continue due to several factors, including global warming and climate change, deforestation, air pollution, and increase of non-airborne environmental pollutants such as plastics, chemicals, and obsolete technology, coupled with insufficient efforts at recycling and the demand for new consumer products. These findings are consistent with studies of climate change risk perception that have found that non-White minorities, specifically Latinx as well as Black Americans, believe in and are concerned about climate change [61]. These individuals have a higher likelihood of living in areas where climate change impacts occur due to discrimination and segregation, which results in increased climate change belief, concern, and response [9,62,63].

Linked to a common belief that heat waves are a consequence of human activity is a common concern about the likely impacts of these climate-induced events on physical and mental health. In this study, perceived risk was based on prior experience with heat waves in addition to information acquired through reading and watching television, formal schooling, and participating in social networks. Moreover, largely based on their own experience, residents of these low-income neighborhoods also asserted that heat waves were likely to make people more depressed, anxious, and irritable. Residents also expressed concern about the health of their children, especially those with chronic health conditions. This concern extended to the ability of schools to serve as safe environments during a heat wave.

### 4.2. Responses to Heat Waves

The second component of our proposed model outlines the adaptive behaviors used by residents of low-income neighborhoods to respond to the risks associated with heatwaves. In addition to knowing to stay hydrated, a wide variety of behaviors are employed to stay cool during a heat wave. Participants almost always mentioned more than one type of behavior they had personally engaged in or recommended for adapting to a heat wave. Several different behaviors were employed simultaneously, including wearing lighter clothing; staying indoors and closing windows, curtains, or blinds; opening windows when temperatures began to drop, going to cooler spaces, eating uncooked foods and ice cream, restricting use of electric appliances, taking frequent showers, communicating with neighbors or extended family, and planting trees on the property for shade. Many of the behaviors practiced were consistent with Centers for Disease Control guidelines, which recommend staying hydrated with water and avoiding sugary beverages, staying cool in an air-conditioned area, and wearing light weight, light-colored, loose-fitting clothes; wearing sunscreen, and avoiding hot and heavy meals [64]. Other behaviors included nutritional practices such as the consumption of certain teas, fruits, and soups by Chinese and Latinx immigrants; restricting use of electrical appliances, and cleaning their places of residence.

Our findings also revealed different levels of response to heat waves, beginning with the level of the individual (e.g., staying hydrated) to the family or household (use of AC, meals provided), organizational level (i.e., school policies and recommendations to students and families), and community level (e.g., checking on neighbors and extended family members). These different levels suggest the inclusion of a socio-ecologic framework [65] in the 4R model to understanding different types of health behaviors utilized and their interrelationships.

### 4.3. Heat Wave Adaptation Resources

Although a majority of study participants possessed AC units within their places of residence and used them during heat waves, one-third of our participants lacked AC either because of restrictions imposed by landlords on the installation of window or wall units, expense of the units themselves or concerns about the threats to health and safety associated with their use. Moreover, consistent with the findings of previous research [9,43], having access to AC does not mean that it is used when needed. In this study, residents often limited use because of concerns about high electric bills. Conversely, residents also reported that use was limited by power outages that tend to occur during heat waves due to excessive use of AC or by warnings from utility companies to limit use due to risk of power outages. Decisions to limit use are also influenced by concerns about the adverse health effects of going from cool, air-conditioned indoor settings to hot, humid outdoor settings. Assessment of AC as a resource for heat wave adaptation must therefore be viewed within the context of ability to use what is available in addition to availability.

Consistent with findings from a primarily low- and moderate- income study sample in a southeastern U.S. city [66], only one-quarter of our study participants reported being contacted by someone in their family or social networks during a heat wave to inquire about their health and well-being. This may be due to average age of these households, as older adults have been reported elsewhere to be the primary beneficiaries of such checkups [66,67]. Additionally, in contrast with previous research documenting limited advanced warning of heat-related events among low-income households [9,43], a large majority of the low-income residents in this study had received advanced warnings from television news and social media.

As with the application of adaptive behaviors at different levels, our findings also revealed different levels of resources to heat waves, beginning with the level of the individual (e.g., knowledge) to the family or household (availability of AC and transportation to cooler spaces), neighborhood or community (e.g., availability of greenspace and social support networks), and the mesoclimate of place of residence. These different levels also suggest the inclusion of a socio-ecologic framework in the 4R model to understanding different types of resources utilized and their interrelationships.

### 4.4. Heat Wave Adaptation Results

In this study, a majority of participants reported one or more physical symptoms of heat exposure, and close to three of every four residents reported mental health symptoms during a heat wave, largely due to uncertainty, confinement, and a sense of desperation. These findings are consistent with studies linking heat waves to increased respiratory and cardiovascular morbidity and mortality as well as suicidal and homicidal behavior [20,21,22,23,24,25] and increased monthly temperatures of 30C and higher to greater mental health difficulties [16]. However, only two of the participants sought medical attention for these problems. Although every participant indicated a willingness to seek medical attention, if necessary, few reported any predisposing medical or psychological conditions that would make them vulnerable to adverse heat-related health effects.

### 4.5. Vulnerability or Resilience

As noted in the introduction, in reaching an agreement about how to model adaptive capacity for responding to heat waves, one of the persistent questions to be addressed concerns the relationship between vulnerability and resilience. In this study, most of the participants reported some physical or mental health problem or symptom during heat waves suggesting vulnerability to heat waves. However, only two residents reported seeking medical care suggesting resilience to heat waves. Almost all participants asserted that heat waves were likely to increase in frequency and intensity with adverse health effects for vulnerable populations. At the same time, they relied on past experience with heat waves and high summer temperatures to guide them in responding to recent heat waves and preparing for future heat wave events. Despite limited resources, low-income residents of urban heat islands utilize a wide range of behaviors to minimize the severity of health impacts, suggesting they are both vulnerable and resilient to heat waves. These findings also suggest that social and environmental vulnerability serve as barriers by limiting the resources available to adapt to heat waves and the ability to utilize those resources in responding to heat waves. Resilience, on the other hand, is a product of successful adaptation and serves to facilitate adaptation by serving as a resource along with the individual and collective knowledge of residents and the supportive infrastructure of their neighborhoods. Hence, both vulnerability and resilience are influenced by adaptation results and, in turn, influence adaptive capacity; it is not a question of either/or but a question of both that determines whether adaptation resources are sufficient to support effective adaptation behaviors.

### 4.6. Limitations

Although the study revealed several novel findings, caution must be exercised in interpretating their implications. This study was based on a small, nonrandom sample of residents in two low-income neighborhoods in Los Angeles, California, whose children were participating in a program designed to prepare them for college; thus, the findings may not be generalizable to residents of similar communities in other urban centers who may have had different levels of exposure to heat waves or access to different resources. Nevertheless, the categorization of communities of residence by median household income, deviations from average outdoor temperatures in other parts of the city, and percent minority status of the community would suggest our findings are likely to be comparable to other urban heat islands with similar characteristics.

Second, all but one of our study participants were female. While gender was previously found to be unrelated to heat risk awareness [32], we have found no studies to date documenting gender differences in heat adaptation. Nevertheless, future studies should include sufficient samples of males for gender-based comparisons.

Third, no formal assessments of physical or mental health status were conducted; assessments of the health effects were based entirely on self-report. Although our findings were consistent with previous research, longitudinal assessments of physical and mental health status of residents in low-income urban heat islands are highly recommended to disaggregate the impact of heat waves from other social and environmental determinants of health.

Finally, because of the small sample size, we performed no statistical evaluations of associations between demographic characteristics, resources, adaptation behaviors, and health outcomes. Although our qualitative findings point to the existence of such associations, they must be verified with larger samples in quantitative studies.

## 5. Conclusions

Despite these limitations, our findings point to the existence of specific patterns of shared knowledge, attitudes, and behaviors among residents of low-income urban heat islands that are grounded in prior experience with heat waves, access to information on the causes and consequences of climate change in general and heat waves in particular, and social networks. Several adaptive behaviors are utilized during heat waves to compensate for lack of AC or limitations to the use of available AC. Other behaviors such as seeking medical attention are widely considered to be options for adaptation but are rarely utilized during a heat wave due to a lack of perceived need. Community support for these multifaceted adaptive strategies is highly recommended and might include financial subsidies and zoning policies to permit purchase an installation of AC units and payment of electricity bills, expansion of green space in such neighborhoods, and school-based programs to ensure student health and safety during these events. Future research is recommended to quantify measures of adaptive capacity, vulnerability, and resilience and examine their associations with standardized measures of physical and mental health outcomes using mixed-effect multivariate methods that model hypothesized relationships between the three sets of determinants of health outcomes on individuals nested within families, urban heat island neighborhoods, and surrounding communities.

## Figures and Tables

**Figure 1 ijerph-19-11090-f001:**
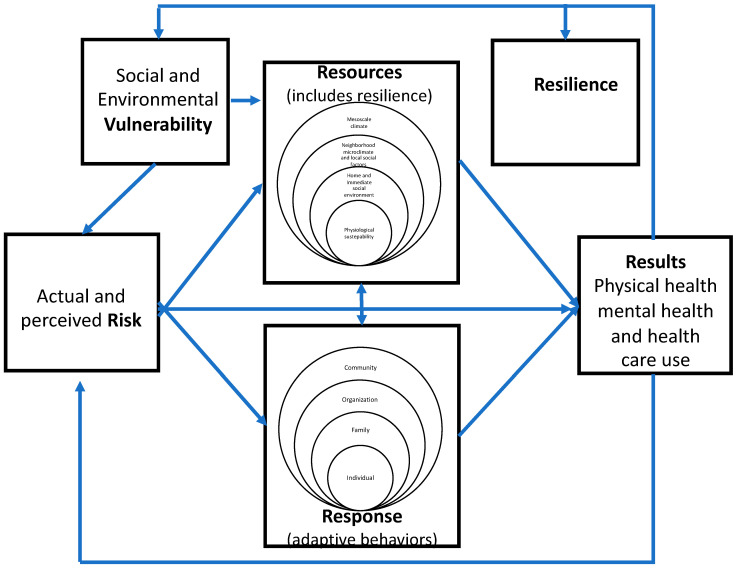
The 4R integrated model of heat wave adaptation.

**Table 1 ijerph-19-11090-t001:** Demographic characteristics and heat adaptation resources of study participants (n = 40).

Characteristic	Mean		S.D.		Range	
Age	42.18		7.4		30–68	
Median household income (USD)	49,358		15,261		30,530–100,866	
Years in neighborhood	14.94		9.57		2–47	
Urban heat index	7229.71		5706.56		1627.1–30,679.8	
Gender	Number			Percentage		
Female	39			97.5		
Male	1			2.5		
Education						
<HS	9			22.5		
HS	16			40.0		
Post-secondary/college	15			37.5		
Employed outside the home	23			57.5		
Race/Ethnicity						
Asian	9			22.5		
Latinx	27			67.5		
African American	1			2.5		
White	3			7.5		
Place of residence						
Own	11			27.5		
Rent	29			72.5		
Air conditioners in home	25			62.5		
Central air conditioning	7			17.5		
Window units	10			25		
Portable units	6			15		
Mini-split units	2			5		
Fans in bedroom	24			60		
Fans in other rooms	7			17.5		
Contact by social network	10			25		
Received warning in advance of heat wave	32			80		

**Table 2 ijerph-19-11090-t002:** Heat wave threat assessment.

Code	N	Illustrative Quotes
Trend in past 10 years	38	“I think in my own opinion that there’s more.”
Reason for increase		
Environmental contaminants	15	“I really don’t know where to recycle my phone, yes there are boxes in various malls and all that, but there are people who say oh, where do I recycle my phone and they throw the batteries in the trash. All that is contaminating since one of the smallest piles contaminates almost 20 L of water. If you throw it in the sea, then. Then I say, wow!”
Global warming/Climate change	10	“Yeah, totally. I mean, it’s global warming. It’s frickin science like um, yeah. Our sea levels are rising, you know, and with all the gases, the greenhouse gases we’re admitting into like our atmosphere now that traps heat in.”
Deforestation and less green space	10	“In the past there were many more trees, even one in each house, but lately they are being cut down. Why? We do not know. Perhaps because the houses are not well given or because. I don’t know yet, because here in our community the trees were previously larger. Lately they are being cut smaller.”
Air pollution	6	“I think that the atmosphere is undergoing changes because of. For all the chemicals. All that pollution that sometimes companies. Or even oneself littering. We are destroying the environment and that is why. That we are already are already feeling it. And I feel that’s why the ozone layer is increasingly sending stronger and stronger sun rays.”
Lack of concern for environment	6	“In other words, sometimes I believe that we are the ones who are destroying our own planet in some way, that we are not taking care of it and that above all we are not teaching our children how to take care of things. So, in that way I think we are affecting the planet, like when sometimes we say why did it get so hot or why is this happening, or why. But we must first start at home. I mean, for example, I didn’t know that eggshells or banana peels could be recycled.”
Overpopulation	5	“You know, there’s a lot of more like people like, everybody is driving in their cars every day and stuff like that, so I feel like all of that, like has an impact on that.”
Wildfires	3	“All the fires have been going on. I feel like there’s been so many more every single year. And it’s just, you know, it’s the pollution is just making it so much more hard.”
Dependence on fossil fuels	2	“I know global warming, I know with the emissions. You know it’s not helping anything. I know specifically with the different things that we’ve done in the classroom that although we are solar and wind and we’re trying to do all these things, we’re still super dependable on fossil fuels.”
New technology	2	“You know, there’s a lot a lot of pollution. There is a lot of new technology that is making it worse. One of those you know that we’re building stuff.”
Anticipated health problems	33	“Physical because if the body is not hydrated, then what is going to happen there? Will you hemorrhage? The kidneys are not going to start working? Internal organs? And that will create health problems in the physical part. In the mental part, since the brain is not sufficiently hydrated, what is going to happen? It is not going to start working. People will have hallucinations and then there will be cases of violence, irritation, dehydration.” “So you see. The elderly and children are the most affected. Ah, yes, heat waves affect them the most. Well, asthma and asthma when people have asthma and the heat, the pollution, all the air, well, that is what goes directly to the lungs and that is what affects them a lot. And then dehydration as well. If you do not hydrate older people or one with children who do not give enough fluid, it can also dehydrate people.”
Concerns about children	13	“Well, my kids sweat a lot. And they get, you know, they get. I guess I call them like they’re extra warm blooded. So, I know I am to worry about if they’re getting water at school, if they’re staying cool, if they’re being provided additional shade at school or they’re being provided cold water or whatnot. So, I do worry about that if I’m not with them.”

**Table 3 ijerph-19-11090-t003:** Heat adaptation behaviors.

Strategy	N	Illustrative Quotes
Stay hydrated	40	“So, we spend the whole day drinking so as not to get dehydrated, because sometimes at night when we sweat we feel that we are dehydrated too. And although we are in the house and all that, but we feel from that not to say we are ok. I drink a lot of water with lemon.”
Seek medical attention	31	“Well, first thing, I think, is make sure I have enough fluids, and if I start feeling dizzy call for help, or call a neighbor or family member. And if it really is bad call 911.”
Go someplace cooler	27	“I take my girls to the park, or they are on swim teams. So, we hardly suffered from being in here and feeling the heat. We spent it in parks. So, wherever you go to get the swimming lesson there is a park and that’s where we spent most of the heat waves.”
Keeping cool	22	“Also, take a bath with cool water, right? If it is very hot, suffocating, it’s like putting them the bathtub and cooling off or being outdoors in the shade.” “We do, I will sometimes like wet paper towels, they can put it on top of their forehead.”
AC use	22	“Well, I think that what we have to do is to prepare ourselves with more water and if we need air conditioning on those days, because sometimes if it is very hot at night, we can’t stand it.”
Limited AC use	19	“We try to save as much energy as possible. So the rule we had to put an air conditioner was that we were only going to put it on when it was already over 90 or when it is hot and we only use it at night from 09:30 to cool down a little and cool the house a little and then we turn it off to sleep later” “Usually, I only use it to cool down and then I turn it off. And no longer because as my husband is very sensitive about it being too cold indoors and the children go outside, and it harms them, and they get sick. So, we almost never wanted to get used to using the air conditioning.”
Avoid going outside	19	“The normal thing is that the same thing that if there’s a lot of heat, to stay inside the house.”
Household modifications	17	“And not using the washing machine, the dryers, the stoves. To make you feel more, cooler and. And we are not expending too much energy.” “Yes, we tried to always ventilate the rooms, open the windows, change the sheets because sometimes it is that the bedspreads are somewhat thick. Put sheets, fresh sheet apart from the windows. The fans, try to clean them just because also they are dusty or not accessible. They also don’t give a lot of air. Also try to have more space, not to have everything saturated, try to have everything clean, sweeping, mopping, because it is also like the floor is cooler too.” “We took different steps, we planted trees around to create a little more shade in the house, but not the trees as the dryness makes them lose hydration and no, you don’t feel any air.”
Clothing	15	“And here in the house when it is very hot, try to be in shorts, in shorts, in a cooler t-shirt.” “I prepared with the children. We take the clothes, because when they are too hot, we wet them and then we put them on wet, or we fill with spray bottles. And I also sprinkled water on my children with fresh water.”
Nutritional practices	15	“That’s when you know you don’t want to stuff yourself with pork belly stew or something like that, right? You want to eat a lot of, you know, smoothies, shakes, and, you know, vegetables and lean meats right?” “No, no, and then also we do a lot of shakes during the summer to keep ourselves cold. And you know store ice cream.” “You know, eat some cool stuff like watermelon and things like that.” “Chinese people like to cook tea with cooling effects. It helps to clear the heat.”

**Table 4 ijerph-19-11090-t004:** Health impacts of heat waves.

Code	N	Illustrative Quotes
Physical health	23	
Fatigue/lethargy	7	“Tired. The heat kind of tires you out. Well, children are lethargic when they are asleep and every now and then because of the heat.”
Headaches	4	“Well, when it’s the hot season, it gives me a lot of headaches.”
Rashes	4	“So for me personally, the effect that the heat had on me was it made my eczema kind of really come out and it was just really uncomfortable.”
Difficulty breathing	2	“Yes, when it’s hot, I feel that my breathing is not very good.”
Loss of appetite	2	“My appetite is not very good.”
Physical discomfort	2	“Okay, so in terms of like your health and well-being it was mostly just physically uncomfortable. You know feeling sweaty and gross…”
Nosebleeds	2	“My girl or my child, what my girl suffers a lot in hot weather is that she bleeds a lot from her nose, she does get much blood in her nose. And then at that time, the times when it was very hot, I was very worried because she was losing a lot of blood, so I went to the doctor with her pediatrician and I told him, look, this is happening, and he told me that it is normal because of the heat wave.”
Dry throat	1	“Yes, there are also four of us. The two children, my husband and me. Yes, there were changes in that time of hot weather. In the throat, that is what affects us the most. Because it is very dry. Very dry, very dry. And our throats get dry.”
Flashes	1	“I got a lot of flashes.”
Cracked lips	1	“Cracked lips were seen in my children and me. So that implied that we needed to hydrate more.”
Swollen feet	1	“Well, you know, when it’s so hot I know my feet is getting swelling.”
Mental health	29	
Anxiety	19	
Due to uncertainty	3	“Especially the anxiety and not knowing what tomorrow is going to be like, whether it is going to be a day when you can stand the heat, or it is going to be a day when you are going to have to be indoors, or it is going to be a day when you have to go somewhere else to feel cool. That gives anxiety. And anxiety also comes to take away sleep, to take away hunger, to take away from focusing on what one should be focused on. So that brings with it many more diseases and now we see more common diabetes, obesity, emotional uncontrol and we think that right now as a mother I was thinking that it is the pandemic, but if we can see with my children, I can see that my child since he was three years old told me ‘Mommy, why, is it so hot?’ And next year it will be like that. So, if my three-year-old is asking me these questions, it is because mentally it is already affecting him.”
Due to confinement	3	“It would make me anxious not to be able to go out.”
Due to concern for children	7	“Not for me personally, but for my children, because they wanted to go outside to play. Then they felt a little frustrated at not being able to go out and play.”
Due to poor health	1	“Um, people with diabetes, because I’m diabetic so, I noticed that, now that I’m turning older, I’m more prone to feeling anxious about the heat. And, especially when there’s high humidity with the heat that that seems to debilitate me.”
Due to desperation	4	“More anxious because, well, the heat does make you desperate.”
Due to poor quality of life	1	“And so I think just the quality of life goes down so much with heat, and especially for people without air conditioning, and people, you know their parents might be more frustrated and not treat them as well, different things like that that.”
Depression	7	“It kind of dampened our spirits with so much heat. There was no desire to do anything.”
Anger/irritability	3	“I don’t know, I never liked it. Always when there is heat, I get more of a temper.”

## Data Availability

The data presented in this study are available on request from the corresponding author.

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
