# Peer review of "Vulnerable, Resilient, or Both? A Qualitative Study of Adaptation Resources and Behaviors to Heat Waves and Health Outcomes of Low-Income Residents of Urban Heat Islands"

_ijerph, 2022, doi:10.3390/ijerph191711090_

Round 1
Reviewer 1 Report
This study is well done from a practical point of view. Research design, data collection, data evaluation method are appropriate and the results presented are in line with the objectives of the article. In general, the focus of this article is more on the practical side, and the outputs are presented mostly for the use of practitioners. Meanwhile, the important part of this article, which is the theoretical aspect, has been neglected. It is true that the structure the article is appropriate, but there is nothing in the text of the article that proves that this article is theoretically important. Therefore, I provide recommendations for the authors below to justify this article scientifically (not just practically), since the journal they are considering is an academic journal, not a news journal.
1- The authors should write a statement of the problem in the introduction section and prove theoretically and scientifically by citing sources/statistics that firstly there is a research problem and secondly this research problem is scientifically worth academic research take place on it.
2- In the introduction section, the authors should show what is the approach of others in the research to address this research problem and justify the innovation of their research.
3- If the introduction and statement of the problem are well written, there is no need for a separate section to describe the literature of the subject, otherwise, the authors should create a separate section after the introduction under the title of research background (or theoretical framework). The final goal of this section should be the development of research questions/hypotheses, and the steps to reach these hypotheses should be presented in this section, relying on the subject literature.
4- In the discussion section of this article, the contributions (not the results) of this article to the literature should also be discussed.
5- In accordance with the theoretical contributions presented in the discussion section, in the conclusion section, the authors should also provide suggestions for future research.
Author Response
This study is well done from a practical point of view. Research design, data collection, data evaluation method are appropriate and the results presented are in line with the objectives of the article. In general, the focus of this article is more on the practical side, and the outputs are presented mostly for the use of practitioners. Meanwhile, the important part of this article, which is the theoretical aspect, has been neglected. It is true that the structure the article is appropriate, but there is nothing in the text of the article that proves that this article is theoretically important. Therefore, I provide recommendations for the authors below to justify this article scientifically (not just practically), since the journal they are considering is an academic journal, not a news journal.
1- The authors should write a statement of the problem in the introduction section and prove theoretically and scientifically by citing sources/statistics that firstly there is a research problem and secondly this research problem is scientifically worth academic research take place on it.
Response: We thank the author for this suggestion. We have included a paragraph in the introduction (lines 54-90) with a definition of adaptive capacity and its relationships to two concepts widely used in examining the impacts of climate change: vulnerability and resilience.
“To prevent the occurrence of these adverse impacts, increased attention has been devoted to reducing social and environmental vulnerability [3, 26,27] and to building, supporting, and sustaining individual and community resilience [28-30]. Resilience is the ability of an individual or community to resist, absorb, accommodate, adapt to, transform and recover from the effects of persistent stress or a disruptive event in a timely and efficient manner [31,32]. Resilience is also related to the concept of adaptive capacity, defined as “the ability of systems, institutions, humans and other organisms to adjust to potential damage, to take advantage of opportunities, or to respond to consequences” [33, p.542]. Adaptive capacity is often linked to vulnerability; however, determining the natures of the relationships involving these two constructs is complicated by differences in how they are defined in the literature and how they are linked together [34]. For instance, in a risk-hazard approach, vulnerability is determined by adaptive capacity, whereas in a contextual approach, vulnerability determines adaptive capacity [35, 36]. Interpreting vulnerability as an outcome leads to a focus on mitigation of greenhouse gas emissions, technical adaptations, and compensation for climate change. In contrast, interpreting vulnerability as contextual results in a focus on addressing existing social and economic marginalization and inequities and adopts social adaptation and sustainable development as the solution [35].
Social and environmental vulnerability to heat waves has been studied extensively [3,5,37,38], especially as it relates to risk perception [ 37, 39-43]. Empirical studies have found that low-income individuals are less likely than higher-income individuals to be aware of heat warnings, but more likely to be concerned that heat would make them ill [37,39-43]. Low-income individuals also less likely to have AC due to its expense, a lack of perceived need, or personal dislike [40--42]. They are less likely to engage in positive adaptive behaviors due to limited resources [40]. However, this research has provided us with only a limited perspective on the adaptive capacity of vulnerable populations. As described by Martin and Paneque [44, p.2],
“Adaptive capacity is a common component of vulerability and resilience assessments, depending on how these concepts are framed, but the isolated measuring and modeling of adaptive capacity is not as advanced as the broader concepts (e.g., vulnerability, resilience). Although several scholars are trying to frame and quantify this ability of adapting to actual and future events, there is still very little consensus as to what constitutes effective and adaptive adaptation, and therefore little agreement about models and indicators to measure it”
In reaching such agreement, one of the persistent questions to be addressed concerns the relationship between vulnerability, viewed as a barrier to adaptive capacity, and resilience, viewed as a facilitator to adaptive capacity.
2- In the introduction section, the authors should show what is the approach of others in the research to address this research problem and justify the innovation of their research.
Response: We explain that our approach to addressing the problem is to conduct a qualitative investigation of the lived experience of heat waves among residents of low-income urban heat islands in the following paragraph (lines 91-102):
“Given the absence of a widely accepted model or theory of adaptive capacity and behaviors, qualitative methods are recommended to explore the parameters and potential causal linkages underlying a phenomenon in order to develop a model that will lead to the generation and testing of hypotheses using quantitative methods [45]. Earlier research has not examined in depth the lived experience of adapting to heat waves of those who are most vulnerable, low-income residents of urban heat islands [11]. Two qualitative studies [43, 46] identified adaptive behaviors, but only those of older adults. An understanding of this experience is critical to identifying resource needs and developing better assessments of adaptive capacity and interventions targeting this population. Qualitative research is also needed to better understand racial/ethnic and socioeconomic disparities in adaptive resources (e.g., AC) and behaviors and sharpen interventions to address root causes [37].”
3- If the introduction and statement of the problem are well written, there is no need for a separate section to describe the literature of the subject, otherwise, the authors should create a separate section after the introduction under the title of research background (or theoretical framework). The final goal of this section should be the development of research questions/hypotheses, and the steps to reach these hypotheses should be presented in this section, relying on the subject literature.
Response: At the suggestion of the reviewer, we have revised the manuscript to clarify our research questions (lines 106-113):
“Our aims were to explore the following questions: 1) How do low-income residents of urban heat islands respond to heat waves, i.e., what adaptive behaviors (adaptation strategies) do they implement during a heat wave? 2) What resources (adaptation capacity) are available to these residents to enable them to adapt to heat waves, i.e., vulnerability and resilience? 3) how are vulnerability and resilience associated with adaptive capacity? and 4) How have such events and the adaptation behaviors and resources implemented in response to them affected the physical and mental health of these residents, i.e., health outcomes?”
However, we also state that as this is an exploratory study, our intent is to develop a conceptual framework that can be used to formulate and test hypotheses. Our use of qualitative data was designed to understand these issues in depth and not to validate an existing framework.
4- In the discussion section of this article, the contributions (not the results) of this article to the literature should also be discussed.
Response: We have revised and reorganized the discussion section, beginning with the introduction of a model of heat wave adaptation (lines 519-524 and Figure 1)
“The four themes identified from the interview transcripts in this study reflect the role of knowledge (a resource), attitudes (perceived risk), and behavior revealed in four major components of an integrated model of heat wave adaptation, illustrated in Figure 1: Risk of heatwaves, Responses or adaptive behaviors, Resources of adaptive capacity, and Results of health impacts of heat waves. The relationships between and among these components are described below:”
We have included a figure to illustrate the four primary components of heat wave adaptation, their relationships to one another, and their relationship to vulnerability and resilience. The remainder of the discussion section highlights how the results illustrate each of the components of the model and their interrelationships.
5- In accordance with the theoretical contributions presented in the discussion section, in the conclusion section, the authors should also provide suggestions for future research.
Response: We now provide suggestions for future research in the conclusion section (lines 924-929).
“Future research is recommended to quantify measures of adaptive capacity, vulnerability and resilience and examine their associations with standardized measures of physical and mental health outcomes using mixed effect multivariate methods that model hypothesized relationships between the three sets of determinants of health outcomes on individuals, nested within families, urban heat island neighborhoods, and surrounding communities.”
Reviewer 2 Report
The manuscript is interesting and pertinent, but it lacks important details:
1.Title: Please make your title more appealing/attractive to maximize the chances of reach and citation; use the main finding as a way out; the use of questions too.
In the current title it is imprecise, and it would be interesting to clarify the registered adaptations.
2. Abstract: Understand your Abstract as a continuation of the title. Originality must be very clear in the abstract, and its conclusion must necessarily answer the objective. In this case, a good way out would be to ask a question in the title and answer it in the Abstract/conclusion...
3. The method is particularly fragile and does not follow international recommendations.
It sticks to a formal description without highlighting how the issue is tackled; the relationship between the study site, the sample and the universe, the data saturation paths and criteria, among other details that provide robustness to the study are absent. There is also no deepening of the research instruments, fieldwork and the way in which the analysis was carried out and subsidized.
4. About the instrument / data collection:
-Who made the instrument
-What is the theoretical framework that supported its construction;
-Describe your variables, type of questions, way of evaluating responses
- Was there a validation or pre-test? If so, how did the instrument's validation process take place (content, clarity and objectivity) and what technique was used for consensus in the questions. This is very important since most of these questions have instruments validated in the literature for this subject. In the absence of a validation process, the research is subject to measurement and interpretation visions
5.Results do not reinforce originality.
The presentation of the results cannot depart from the simple description of the empirical data, except for a formal categorization that is not explained. What criteria were used to select the speeches? What is the process of making speeches?
Author Response
Please note: The lines mentioned in our responses may not correspond to the text that has been uploaded. For some reason, we have noticed changes in lines with each version saved on Word
Title: Please make your title more appealing/attractive to maximize the chances of reach and citation; use the main finding as a way out; the use of questions too.
In the current title it is imprecise, and it would be interesting to clarify the registered adaptations.
Response: At the suggestion of the reviewer, we have revised the title, framing it as a question. The title is now as follows:
“Vulnerable, Resilient, or Both? A Qualitative Study of Adaptation Resources and Behaviors to Heat Waves and Health Outcomes of Low-income Residents of Urban Heat Islands”
- Abstract: Understand your Abstract as a continuation of the title. Originality must be very clear in the abstract, and its conclusion must necessarily answer the objective. In this case, a good way out would be to ask a question in the title and answer it in the Abstract/conclusion...
Response: At the suggestion of the reviewer, we have revised the abstract so that it answers the question in the title of the article (lines 25-30).
- The method is particularly fragile and does not follow international recommendations.
Response: We have included the following statement at the beginning of the Methods Section (lines 115-116). We also provided a completed SRQR form as a supplementary material.
“Authors followed the Standards for Reporting Qualitative Research (SRQR) to report the findings from this study [47] (see Supplementary Materials).”
It sticks to a formal description without highlighting how the issue is tackled; the relationship between the study site, the sample and the universe, the data saturation paths and criteria, among other details that provide robustness to the study are absent. There is also no deepening of the research instruments, fieldwork and the way in which the analysis was carried out and subsidized.
Response: We provide more detail on sample selection criteria (lines 129, 194-197), the questions asked and the rationale for asking them (lines 211-215), determination of data saturation (lines 198-200), and the process of data analysis (lines 251-283). We are unclear what the reviewer means by the way in which the analysis was subsidized. Do you mean how it was financed? We note in the discussion section that the study sample was not intended to be representative of all residents of all low-income urban heat islands, hence caution must be exercised in interpreting findings (lines 720-729). We recommend that future research involve a quantitative investigation of a larger, more representative sample living in different urban areas (lines 924-929).
- About the instrument / data collection:
-Who made the instrument
Response: The interview guide was developed by the investigators
-What is the theoretical framework that supported its construction;
Response: We now describe that we employed an Integrated Climate Change and Health Indicator Systems Framework developed by Liu and colleagues (2021) (lines 211-213).
.
“Using a semi-structured interview guide we developed that was based on the Integrated Climate Change and Health Indicator Systems Framework34, all interviews were conducted online using the Zoom platform or by telephone.”
-Describe your variables, type of questions, way of evaluating responses
Response: We provide more detail on the questions asked, how answers were confirmed, and how the data were analyzed. We have also provided the interview guide as one of the Supplementary Materials.
“Interviewers summarized participant responses to questions throughout the interview to ensure their accuracy and validity.”
- Was there a validation or pre-test? If so, how did the instrument's validation process take place (content, clarity and objectivity) and what technique was used for consensus in the questions. This is very important since most of these questions have instruments validated in the literature for this subject. In the absence of a validation process, the research is subject to measurement and interpretation visions
Response: The reviewer is correct in stating that there exist several validated instruments on this subject in the literature. However, our semi-structured interviews were designed to explore the lived experience of coping with extreme weather events and to obtain an in-depth understanding of these experiences, not to validate pre-conceived ideas or test hypotheses. Therefore, we did not rely of the use of validated instruments. As we note in the methods section, consensus among participants was estimated on the basis of the frequency of similar responses to the questions that were asked during the interviews (lines 281-283).
- Results do not reinforce originality.
The presentation of the results cannot depart from the simple description of the empirical data, except for a formal categorization that is not explained. What criteria were used to select the speeches? What is the process of making speeches?
Response: We are not entirely certain of what the reviewer means by the results “cannot” depart from the simple description of the empirical data” and by the use of the term “speeches.” We have revised and reorganized the discussion section to highlight the implications of our findings for understanding the relationships between adaptive capacity, vulnerability and resilience and their associations with health outcomes. This revision includes the introduction of a model of heat wave adaptation (lines 519-524 and Figure 1).
“The four themes identified from the interview transcripts in this study reflect the role of knowledge (a resource), attitudes (perceived risk), and behavior revealed in four major components of an integrated model of heat wave adaptation, illustrated in Figure 1: Risk of heatwaves, Responses or adaptive behaviors, Resources of adaptive capacity, and Results of health impacts of heat waves. The relationships between and among these components are described below:”
We have included a figure to illustrate the four primary components of heat wave adaptation, their relationships to one another, and their relationship to vulnerability and resilience. The remainder of the discussion section highlights how the results illustrate each of the components of the model and their interrelationships.
If the reviewer means by speeches the interviews that were conducted, we explained the process of participant recruitment and selection and how the interviews were conducted in the methods section.
Round 2
Reviewer 1 Report
The authors have revised the paper according to my comments and all of my concerns are considered. Therefore, I recommend this article to be considered for publication.
Reviewer 2 Report
The text has been significantly improved